# CASPR: Combining Axes Preconditioners through Kronecker Approximation for Deep Learning

**Sai Surya Duvvuri**[*]
Department of Computer Science
The University of Texas at Austin
saisurya@cs.utexas.edu

**Fnu Devvrit**
Department of Computer Science
The University of Texas at Austin
devvrit@cs.utexas.edu

**Rohan Anil**
Google DeepMind
rohananil@google.com

**Cho-Jui Hsieh**
CS Department, UCLA & Google
chohsieh@cs.ucla.edu

**Inderjit S. Dhillon**
Google
isd@google.com

## Abstract

Adaptive regularization based optimization methods such as full-matrix Adagrad which use gradient second-moment information hold significant potential for fast convergence in deep neural network (DNN) training, but are memory intensive and computationally demanding for large neural nets. We develop a technique called Combining AxeS PReconditioners (CASPR), which optimizes matrix-shaped DNN parameters by finding different preconditioners for each mode/axis of the parameter and combining them using a Kronecker-sum based approximation. The Kronecker-sum based combination allows us to show that CASPR is ordered between a well-known Kronecker product based combination, Shampoo, and full-matrix Adagrad preconditioners in Loewner order, as a result, it is nearer to full-matrix Adagrad than Shampoo. We also show tighter convergence guarantees in stochastic optimization compared to Shampoo. Furthermore, our experiments demonstrates that CASPR approximates the gradient second-moment matrix in full-matrix Adagrad more accurately, and shows significant improvement in training and generalization performance compared to existing practical adaptive regularization based methods such as Shampoo and Adam in a variety of tasks including graph neural network on OGBG-molpcba, Transformer on a universal dependencies dataset and auto-regressive large language modeling on C4 dataset.

## 1 Introduction

Adaptive methods, including Adagrad (Duchi et al., 2011) and its variants such as Adam (Kingma & Ba, 2014) and RMSProp (Tieleman & Hinton, 2012), are widely employed for training neural networks. These approaches calculate updates by scaling each coordinate of the gradient direction (or momentum in case of Adam) $(g_t)_i$ with an adaptive learning rate, which is tuned using the second moments $\sum_t (g_t)_i^2$ of the gradients before updating the weights. This process is equivalent to pre-multiplying the gradient by a diagonal matrix (preconditioner) to perform the weight update. On the other hand, full-matrix Adagrad (Duchi et al., 2011) stores cross-moments, i.e., the $(i, j)$-th element of the (full-matrix statistic) $\sum_t g_t g_t^\top$, where $g_t \in \mathbb{R}^d$. However, this can be infeasible for modern neural networks with large number of parameters due to the $\mathcal{O}(d^2)$ memory requirement to store the second-moments and $\mathcal{O}(d^3)$ computational complexity to perform inverse square root operations.

We address this issue by first proposing a set of preconditioners- *axes preconditioners*, one for each axis/dimension of matrix shaped parameters in a deep neural network, which is derived by finding a block-diagonal approximation with identical blocks such that it is nearest to full-matrix

---

[*]Work done as a student researcher at Google

statistic. Each approximation offers the advantages of a) fast inverse square root operation (which then gives the axes preconditioner), and b) low memory. However, using only one axes preconditioner throughout the training can be suboptimal, as it only stores partial information about full-matrix statistic. To this end, we first propose a sequence of combinations of axes preconditioners using Kronecker-sums, and subsequently show that a Kronecker product based preconditioner (an existing preconditioner - Shampoo (Anil et al., 2020; Gupta et al., 2018)) to be at one end of this sequence, we find our proposed novel combination - CASPR to be at the other end of the sequence (Lemma 3.2). CASPR preconditioner is carefully picked from the proposed set of combinations such that it is ordered (sandwiched) between Shampoo and full-matrix Adagrad in Loewner order (Löwner, 1934), as a result, it is nearer to the more powerful full-matrix Adagrad preconditioner than Shampoo in Loewner order. This is acheived by utilizing the simple inequality in Loewner order: $(A + B)^2 \succeq 4AB$ for commutable and positive definite $A$ and $B$. Surprisingly, we show that the established Loewner order also helps us derive tighter regret bounds in the online convex optimization framework (Hazan et al., 2016; Shalev-Shwartz et al., 2012) than Shampoo, which translates to convergence guarantees in stochastic convex optimization via online-to-batch conversions (Cesa-Bianchi et al., 2004). Furthermore, we conduct numerical experiments to measure nearness of the CASPR approximation to full-matrix statistics encountered during deep neural network training and show that our CASPR approximates full-matrix statistics better than individual block-diagonal approximations (with identical blocks) corresponding to each axis preconditioner and Shampoo.

**Contributions**: We develop a practical adaptive regularization method which uses a Kronecker-sum based combination of axes preconditioners. We prove a tighter regret bound than a well-known practical adaptive method - Shampoo (Gupta et al., 2018; Anil et al., 2020) in online convex optimization framework. Note that regret bounds in online convex optimization framework have reductions to convergence guarantee in stochastic convex optimization (Cesa-Bianchi et al., 2004), non-smooth non-convex optimization (Cutkosky et al., 2023) and smooth non-convex optimization guarantees (Agarwal et al., 2019). We conduct numerical experiments to compare the approximation error with respect to full-matrix statistic and demonstrate better approximation error than Shampoo, diagonal preconditioners, and individual axes preconditioners. We train graph neural network (Battaglia et al., 2018) on OGBG-molpcba dataset (Hu et al., 2020), where CASPR gives a relative improvement of $1.4\%$ in test average precision compared with Shampoo and $7\%$ improvement with respect to AdamW. We also train a Transformer network (Vaswani et al., 2017) on a universal dependencies dataset (Nivre et al., 2020), where we observe $1\%$ relative improvement in validation error compared to Shampoo and $3\%$ relative improvement compared to AdamW. To demonstrate scalability of our method we train a 14 million parameter on $\sim 167$ billion tokens and 234 million parameter model with $\sim 42$ billion tokens on C4 dataset (Raffel et al., 2020), where we noticed upto 1% relative improvement in log perplexity. Furthermore, it is worth noting that the simplicity of CASPR would only require a small modification to existing implementations of Shampoo. As a result, we notice that Shampoo and CASPR take almost similar training time in our experiments.

**Notation and Standard Matrix Identities.** We use $A \otimes B$ to denote Kronecker product (Petersen et al., 2008), and $\overline{\text{vec}}(A)$ to denote row-major order vectorization of matrix $A$. Properties of Kronecker product: a) $(A \otimes B^T)\overline{\text{vec}}(G) = \overline{\text{vec}}(AGB)$ b) $(A \otimes B)(C \otimes D) = (AC) \otimes (BD)$. We use Loewner order $A \succeq B$ to denote that $A - B$ is a positive semi-definite (PSD) matrix. We use block-diag$(A_1, \ldots, A_m)$ to denote a block-diagonal matrix with blocks $\{A_1, \ldots, A_m\}$. On the other hand, $\text{diag}(A)_{i,j} = A_{i,i}$ if $i = j$ else 0, preserving only the diagonal entries of $A$. We use $\text{tr}(A)$ to denote trace of a matrix and use $\|A\|_F = \sqrt{\text{tr}(A^\top A)}$ to denote Frobenius norm, and $\langle u, v \rangle = u^\top v$ to denote dot-product between vectors $u$ and $v$. For $A \succeq 0$, $A^{1/p}$ denotes the matrix $p^{th}$ root of $A$.

## 2   RELATED WORK

**Hessian-based methods.** Newton's method is a second-order optimization technique that premultiplies the gradient by the inverse of Hessian, which constitutes second-order partial derivatives to update the model parameters. Although it can have fast convergence properties, it is computationally expensive and inapplicable for large-scale neural networks due to the high computational and memory complexity involved in calculating the Hessian matrix. BFGS (Broyden-Fletcher-Goldfarb-Shanno) (Broyden, 1967; Fletcher, 1970; Shanno, 1970; Goldfarb, 1970) and LBFGS (Limited-memory BFGS) (Liu & Nocedal, 1989) are quasi-Newton methods that implicitly approximate the Hessian matrix, without computing second-order partial derivatives. However, directly applying LBFGS on

neural networks can be infeasible due to high memory requirement in maintaining the gradients of previous iterations in memory.

**Adaptive regularization methods.** Adaptive regularization introduced in Duchi et al. (2011) includes a diagonal preconditioner (diagonal Adagrad) that adapts the learning rate for each parameter individually based on the historical sum of squared gradients. Although full-matrix version of this preconditioner is outlined in this work it has been disregarded for its high memory and computational requirements. There exist low-rank approximations of the full-matrix statistic such as Agarwal et al. (2019), which demonstrate better performance in anisotropic loss landscapes compared to diagonal preconditioners. However, this method requires storing hundreds of gradients from previous iterations, which is memory-intensive. Alternatively, Kronecker product based approximation such as Shampoo (Gupta et al., 2018) reduces the time and memory complexity by leveraging the matrix structure of the gradient. Shampoo, which admits a regret bound in the online convex optimization (OCO) framework (Hazan et al., 2016), has demonstrated superior performance than diagonal preconditioners such as AdamW, RMSProp in large-batch settings (Anil et al., 2020) where the time to conduct large batch forward-backward propagation is high compared to the costly optimizer update rule and also has shown adoption in industry-scale clickthrough rate prediction models (Anil et al., 2022). It is worth noting that regret upper bound guarantees can be converted to stochastic optimization convergence rate guarantees to reach stationary point for smooth non-convex objectives as in (Agarwal et al., 2019) and more recently, conversion to non-smooth non-convex objectives (Cutkosky et al., 2023).

**Fisher approximation to Hessian.** Natural Gradient Descent (Amari, 1998) is an optimization method that uses inverse of Fisher information matrix of the model's predictive distribution as preconditioner. This has shown to be generalized Gauss-Newton approximation of the Hessian matrix (Martens, 2020). Natural Gradient Descent (NGD) suffers from high complexity due to the need to compute the Fisher matrix and its inverse, which can be infeasible to compute for modern neural networks. Kronecker product approximations such as K-FAC (Martens & Grosse, 2015; Ren & Goldfarb, 2021), have appeared to address the complexity issue. However, the gradients used to approximate the Fisher matrix require an additional backpropagation step through a different loss function, which can further increase the computational complexity.

## 3 KRONECKER SUM BASED PRECONDITIONER

Full-matrix Adagrad Duchi et al. (2011); Agarwal et al. (2019) maintains $H_t \in \mathbb{R}^{d \times d}$ and updates the parameters $w_t \in \mathbb{R}^d$ using the following steps:

$$H_t := H_{t-1} + g_t g_t^\top \tag{1}$$

$$w_{t+1} := w_t - \eta_t H_t^{-1/2} g_t, \tag{2}$$

where $f_t \colon \mathbb{R}^{d \times d} \to \mathbb{R}$ is the loss induced by the mini-batch at iteration $t \in \{1, \dots, T\}$, and $g_t = \nabla f_t(w_t)$. Here $H_t$ stores all gradient cross-moments $(H_t)_{i,j} = \langle g_{1:t}^{(i)}, g_{1:t}^{(j)} \rangle$, where $g_{1:t}^{(i)} = ((g_1)_i, \dots, (g_t)_i)$. Storing the entire *full-matrix statistic* $H_t \in \mathbb{R}^{d \times d}$ is infeasible for a deep neural-network (DNN) as $d$ can be in the millions. Furthermore, the inverse square root takes $\mathcal{O}(d^3)$ floating point operations (flops).

To address these issues, we aim to find an approximation $\hat{H}_t$ of $H_t$ that: a) offers a fast inverse operation, b) has low memory representation, and c) approximates $H_t$ well. To enforce these constraints formally, we introduce an optimization sub-problem to determine a sparse approximate statistic $\hat{H}$ constrained to a set of sparse matrices $\mathcal{S}$, before conducting the inverse root to find the preconditioner $\hat{H}_t^{-1/2}$ and updating $w_t$:

$$\hat{H}_t := \arg \min_{\hat{H} \in \mathcal{S}} \|\hat{H} - H_t\|_F, \quad w_{t+1} := w_t - \eta_t \hat{H}_t^{-1/2} g_t. \tag{3}$$

Constraining $\hat{H}$ to the set of all positive semi-definite (PSD) matrices $\mathcal{S} = \{A \in \mathbb{R}^{d \times d} : A \succeq 0\}$ in subproblem (3) trivially gives the full-matrix statistic $\hat{H}_t = H_t$, since $H_t \succeq 0$. Note that inverse square root $\hat{H}_t^{-1/2}$ cannot be conducted on an indefinite $\hat{H}_t$. But, if we choose $\mathcal{S}$ to be the set of diagonal PSD matrices, $\mathcal{S} = \{A \in \mathbb{R}^{d \times d} : A \succeq 0, A_{i,j} = 0, i \neq j \in [d]\}$, the optimal solution is $\hat{H}_t = \text{diag}(H_t)$, which leads to the commonly used diagonal Adagrad update:

$$\hat{H}_t := \hat{H}_{t-1} + \text{diag}(g_t g_t^\top), \quad w_{t+1} := w_t - \eta_t \hat{H}_t^{-1/2} g_t.$$

where, $\text{diag}(A)_{i,j} = A_{i,j}$ if $i = j$, and 0 if $i \neq j$. Maintaining the above $\hat{H}_t$ requires only keeping $d$ diagonal elements of $H_t$ in memory, and $\mathcal{O}(d)$ computations for the inverse square root operation. Nonetheless, the approximation error $\|\hat{H} - H_t\|_F$ can be substantial when using the diagonal sparsity structure since off-diagonal elements of $H_t$ are neglected.

This calls for more accurate approximations $\hat{H}_t$ that also capture cross-moments of gradients. Note that gradients of individual layers in most DNNs assume a matrix structure, for example, weight matrices of fully-connected layers in DNNs. The gradient corresponding to weight matrix $W \in \mathbb{R}^{m \times n}$ can be written as $G_t \in \mathbb{R}^{m \times n}$, with $g_t = \overline{\text{vec}}(G_t)$, where $\overline{\text{vec}}(A) \in \mathbb{R}^{mn}$ denotes a vector with elements of matrix $A$ in row-major order and total number of elements $d = mn$,. For such matrix-structured gradients, we now generalize the diagonal sparsity structure mentioned earlier to the following block-diagonal structure, $\mathcal{S}_R = \{\text{block-diag}(R_1, R_2 \ldots, R_m) : R_i \succeq 0 \in \mathbb{R}^{n \times n}, \forall i \in [m]\}$, where individual blocks $R_i$ capture gradient second-moments within row $i$ of $G_t$. This is related to unit-wise preconditioning in Osawa et al. (2023). Setting $\mathcal{S} := \mathcal{S}_R$ in (3) results in:

$$\min_{\hat{H} \in \mathcal{S}_R} \|\hat{H} - H_t\|_F^2 = \min_{\{R_1,\ldots,R_m\}} \sum_{i \in [m]} \|R_i - H_t^{(i)}\|_F^2 = \sum_{i \in [m]} \min_{R_i} \|R_i - H_t^{(i)}\|_F^2$$

$$\implies R_i^* = H_t^{(i)} = \sum_{s=1}^{t} g_s^{(i)} g_s^{(i)\top}, \forall i \in [m]$$

where $g_s^{(i)}$ denotes the $i$-th row of $G_s$, and $H_t^{(i)} \in \mathbb{R}^{n \times n}$ is the $i$-th diagonal block of $H_t = \sum_{s=1}^{t} g_s g_s^\top$. However, storing all the $R_i$ requires $mn^2$ memory, which can still be impractical when $m$ and $n$ are high. Furthermore, the inverse root operation would require $\mathcal{O}(mn^3)$ flops, as the inverse needs to be computed for each block individually. To alleviate this issue, we propose a simpler block-diagonal sparsity constraint $\mathcal{S}_{I \otimes R}$, where all the $R_i$ are identical, in the following lemma.

**Lemma 3.1.** *Consider the sparsity constraint $\mathcal{S}_{I \otimes R} = \{\text{block-diag}(R, R \ldots, R) : R \succeq 0 \in \mathbb{R}^{n \times n}\}$ in (3), then the optimal solution $R^*$ for (3) (rewritten below) is as follows:*

$$R^* = \arg\min_{R \succeq 0} \|I_m \otimes R - \sum_{s=1}^{t} g_s g_s^\top\|_F^2 = \frac{1}{m} \sum_{s=1}^{t} G_s^\top G_s,$$

*where $g_s = \overline{\text{vec}}(G_s)$.*

*Proof.* Let $H_t^{(i)} \in \mathbb{R}^{n \times n}$ be the $i$-th diagonal block of $H_t = \sum_{s=1}^{t} g_s g_s^\top$. Then:

$$\min_{\hat{H} \in \mathcal{S}_{I \otimes R}} \|\hat{H} - H_t\|_F^2 = \min_{R \succeq 0} \sum_{i \in [m]} \|R - H_t^{(i)}\|_F^2 = \min_{R \succeq 0} m\|R\|_F^2 - \sum_{i \in [m]} \text{tr}(H_t^{(i)} R).$$

$$\implies R^* = \frac{1}{m} \sum_{i=1}^{m} H_t^{(i)} = \frac{1}{m} \sum_{i=1}^{m} \sum_{s=1}^{t} g_s^{(i)} g_s^{(i)\top} = \frac{1}{m} \sum_{s=1}^{t} G_s^\top G_s$$

Similarly, for column preconditioner corresponding to $g_t = \overline{\text{vec}}(G_t^\top)$, defining the sparsity constraint $\mathcal{S}_{L \otimes I} = \{L \otimes I_n : L \succeq 0 \in \mathbb{R}^{m \times m}\}$ and solving the subproblem (3) gives $L^* = \frac{1}{n} \sum_{s=1}^{t} G_s G_s^\top$ (see Appendix A.1, Lemma A.1).

Individually $A = L^* \otimes I_n$ and $B = I_m \otimes R^*$ are optimal approximations for the sparsity constraints $\mathcal{S}_{L \otimes I}$ and $\mathcal{S}_{I \otimes R}$ respectively for the subproblem (3). However, using the corresponding preconditioners $A^{-1/2} = (L^*)^{-1/2} \otimes I_n$ and $B^{-1/2} = I_m \otimes (R^*)^{-1/2}$ (via (3)) individually can be inefficient as they do not approximate gradient cross-moments outside the block-diagonals of the full-matrix statistic $H_t$. To this end, several works have used Kronecker product structure (Martens & Grosse, 2015; Ren et al., 2021; Gupta et al., 2018) for preconditioner approximation. However, we take an alternative approach by the following general combination of axes preconditioners.

**Lemma 3.2** (A general combination of axes preconditioners)**.** *Consider a sequence of preconditioners for $p \geq 1$:*

$$X_t^{caspr}(p) = ((\tilde{L}_t^{-1/4p} \otimes I_n + I_m \otimes \tilde{R}_t^{-1/4p})/2)^{2p},$$

*where, $\tilde{L}_t = \sum_{s=1}^{t} G_s G_s^\top + \epsilon I_m$, $\tilde{R}_t = \sum_{s=1}^{t} G_s^\top G_s + \epsilon I_n$, then*

$$\lim_{p \to \infty} X_t^{caspr}(p) = \tilde{L}_t^{-1/4} \otimes \tilde{R}_t^{-1/4} = X_t^{sh}$$

where, $X_t^{sh}$ is the Shampoo preconditioner, which is an existing Kronecker product based preconditioner (Gupta et al., 2018; Anil et al., 2020).

Lemma 3.2 shows that the Shampoo preconditioner is a limiting case of the general CASPR preconditioner $X_t^{caspr}(p)$. The proof of Lemma 3.2 is given in Appendix A.1. To compare with full-matrix preconditioner, we show in the following lemma, that $X_t^{caspr}(1)$ is nearer to full-matrix preconditioner than the Shampoo preconditioner in Loewner order (Löwner, 1934; Bhatia, 2009). We will use this fact to prove tighter convergence guarantees than Shampoo in Section 4.

**Lemma 3.3** (CASPR preconditioner). *Let* $\tilde{L}_t = \sum_{s=1}^t G_s G_s^\top + \epsilon I_m$, $\tilde{R}_t = \sum_{s=1}^t G_s^\top G_s + \epsilon I_n$ *and* $r = \max_t \text{rank}(G_t)$, *then the CASPR preconditioner* $((\tilde{L}_t^{-1/4} \otimes I_n + I_m \otimes \tilde{R}_t^{-1/4})/2)^2$ *(Algorithm 1) and Shampoo preconditioner* $\tilde{L}_t^{-1/4} \otimes \tilde{R}_t^{-1/4}$ *follow the Loewner order:*

$$\sqrt{r}(r\epsilon I_d + \sum_{t=1}^T g_t g_t^\top)^{-1/2} \succeq ((\tilde{L}_t^{-1/4} \otimes I_n + I_m \otimes \tilde{R}_t^{-1/4})/2)^2 \succeq \tilde{L}_t^{-1/4} \otimes \tilde{R}_t^{-1/4}$$

We give the proof of this lemma in Appendix A.1. Shampoo preconditioner is the geometric mean of $X_t^L = \tilde{L}_t^{-1/2} \otimes I_n$ and $X_t^R = I_m \otimes \tilde{R}_t^{-1/2}$, $X_t^{sh} = (X_t^L X_t^R)^{1/2} = L_t^{-1/4} \otimes R_t^{-1/4}$, where last equality is due to $(A \otimes B)^{1/2} = A^{1/2} \otimes B^{1/2}$ (Petersen et al., 2008) for positive semidefinite $A$ and $B$. This geometric mean interpretation explains the occurrence of exponent $-1/4$ for $\tilde{L}_t$ and $\tilde{R}_t$ in Shampoo preconditioner. The proposed preconditioner $X_t^{caspr}(1)$ can be rewritten as follows:

$$((\tilde{L}^{-1/4} \otimes I_n + I_m \otimes \tilde{R}_t^{-1/4})/2)^2 = \frac{(\tilde{L}^{-1/2} \otimes I_n + I_m \otimes \tilde{R}^{-1/2})/2 + \tilde{L}_t^{-1/4} \otimes \tilde{R}_t^{-1/4}}{2}$$

$$= \frac{(X_t^L + X_t^R)/2 + (X_t^L X_t^R)^{1/2}}{2}.$$

Using the last equality, $X_t^{caspr}(1)$ can be interpreted as the average of arithmetic mean and geometric mean (which is Shampoo preconditioner) of $X_t^L$ and $X_t^R$.

Using the CASPR preconditioner to update the parameter $w_t = \overline{\text{vec}}(W_t) \in \mathbb{R}^d$, where $d = mn$, will give the following update rule:

$$X_t := \left((\tilde{L}_t^{-1/4} \otimes I_n + I_m \otimes \tilde{R}_t^{-1/4})/2\right)^2; \quad w_t := w_{t-1} - \eta X_t g_t \qquad \text{(vector update)},$$

where $g_t = \overline{\text{vec}}(G_t) \in \mathbb{R}^d$. Here we dropped the $\sqrt{r}$ scale factor from Lemma 3.3 as it can be absorbed in the learning rate parameter $\eta$. Forming the entire CASPR preconditioner $X_t \in \mathbb{R}^{d \times d}$ is infeasible, since $d$ can be very large and computing the update $X_t g_t$ would cost $\mathcal{O}(d^2)$. However, expanding $X_t$ and using the identity $(A \otimes C)\overline{\text{vec}}(B) = \overline{\text{vec}}(ABC^\top)$ will give the following feasible update rule:

$$W_{t+1} := W_t - \eta \left(\tilde{L}_t^{-1/2} G_t + 2\tilde{L}_t^{-1/4} G_t \tilde{R}_t^{-1/4} + G_t \tilde{R}_t^{-1/2}\right)/4 \qquad \text{(matrix update)}.$$

This update rule can be further simplified into two preconditioning steps which only requires computing $\tilde{L}_t^{-1/4}$ and $\tilde{R}_t^{-1/4}$ as follows:

$$U_t := \tilde{L}_t^{-1/4} G_t + G_t \tilde{R}_t^{-1/4}$$
$$U_t := \tilde{L}_t^{-1/4} U_t + U_t \tilde{R}_t^{-1/4} \qquad \text{(CASPR update)}$$
$$W_t := W_{t-1} - \eta U_t$$

which performs the above at every iteration by maintaining $L_t$ and $R_t$ in an online fashion (see line 4), which require storing a total of $(m^2 + n^2)$ elements of matrices $L_t$ and $R_t$ in memory. Given the simplicity of the update, it is easy to change the existing implementations of Shampoo to implement CASPR, since both CASPR and Shampoo compute the same axes preconditioners $\tilde{L}_t^{-1/4}$ and $\tilde{R}_t^{-1/4}$, however Shampoo preconditions the gradient $G_t$ differently:

$$U_t := \tilde{L}_t^{-1/4} G_t$$
$$U_t := U_t \tilde{R}_t^{-1/4} \qquad \text{(Shampoo update)}$$

---

**Algorithm 1** CASPR Algorithm

---

1: $W_1 = 0 \in \mathbb{R}^{m \times n}, L_0 = 0, R_0 = 0$
2: **for** $t \in 1, \ldots, T$ **do**
3:    Compute gradient $G_t = \nabla f_t(W_t) \in \mathbb{R}^{m \times n}$
4:    Update preconditioners:

$$L_t := L_{t-1} + G_t G_t^\top, \quad R_t := R_{t-1} + G_t^\top G_t$$

5:    Precondition gradient and update parameters:

$$\text{Compute } \tilde{L}_t^{-1/4} := (L_t + \epsilon I_m)^{-1/4}, \tilde{R}_t^{-1/4} := (R_t + \epsilon I_n)^{-1/4}$$
$$U_t := \tilde{L}_t^{-1/4} G_t + G_t \tilde{R}_t^{-1/4}$$
$$U_t := \tilde{L}_t^{-1/4} U_t + U_t \tilde{R}_t^{-1/4}$$
$$W_t := W_{t-1} - \eta U_t$$

6: **end for**

---

In line 5, the preconditioning of the columns and rows of $G_t$ is achieved sequentially using $L_t^{-1/4}$ and $R_t^{-1/4}$. It is worth noting that this step takes $\mathcal{O}(m^3 + n^3)$ flops due to the inverse operation, which is the same cost incurred by Kronecker product based approaches.

### 3.1 COMPARISON OF CASPR AND SHAMPOO APPROXIMATIONS

Let $X_t^{caspr} = ((L_t^{-1/4} \otimes I_n + I_m \otimes R_t^{-1/4})/2)^2$ and $X_t^{sh} = L_t^{-1/4} \otimes R_t^{-1/4}$ denote the preconditioners of CASPR and Shampoo respectively as in Lemma 3.3 (here $\epsilon$ is dropped for brevity), where $L_t = \sum_{s=1}^t G_s G_s^\top$ and $R_t = \sum_{s=1}^t G_s^\top G_s$, then the corresponding induced approximations $\hat{H}_t^{caspr}$ and $\hat{H}_t^{sh}$ for $H_t = \sum_t g_t g_t^\top$ will satisfy $X_t^{caspr} = (\hat{H}_t^{caspr})^{-1/2}$ and $X_t^{sh} = (\hat{H}_t^{sh})^{-1/2}$ respectively. This relationship is evidenced in the update rule for Adagrad, as in (2). Then the explicit expressions for the approximations $\hat{H}^{caspr}$ and $\hat{H}^{sh}$ and their eigendecompositions is as follows:

**Lemma 3.4** (eigendecomposition of CASPR preconditioner). *Let the eigenpairs of $L_t$ be $\{(\lambda_1, u_1), \ldots, (\lambda_m, u_m)\}$ and $R_t$ be $\{(\sigma_1, v_1), \ldots, (\sigma_n, v_n)\}$, then the eigenpairs of $\hat{H}_t^{caspr} = (X_t^{caspr})^{-2}((L_t^{-1/4} \otimes I_n + I_m \otimes R_t^{-1/4})/2)^{-4}$ and $\hat{H}_t^{sh} = (X_t^{sh})^{-2} = L_t^{1/2} \otimes R_t^{1/2}$ are $\{(((\lambda_i^{-1/4} + \sigma_j^{-1/4})/2)^{-4}, u_i \otimes v_j) : i, j \in [m], [n]\}$ and $\{((\lambda_i \sigma_j)^{1/2}, u_i \otimes v_j) : i, j \in [m], [n]\}$ respectively.*

The proof is given in Appendix A.1, which uses the Kronecker product structure of $\hat{H}_t^{sh}$ and Kronecker sum structure of $\hat{H}_t^{caspr}$ to derive the eigenvalues and eigenvectors. Lemma 3.4 shows that both $\hat{H}_t^{sh}$ and $\hat{H}_t^{caspr}$ possess the same set of eigenvectors, but the corresponding eigenvalues $(\lambda_i^{-1/4} + \sigma_j^{-1/4})^{-4}/4$ and $(\lambda_i \sigma_j)^{1/2}$ are significantly different.

While Lemma 3.4 conveys discrepancies between each eigenvalues of Shampoo and CASPR, we measure overall approximation errors of Shampoo and CASPR with respect to $H_t$ in Figure 1, where CASPR demonstrates better approximation quality than Shampoo.

## 4 CONVERGENCE GUARANTEES

We set up our convergence analysis in the online convex optimization (OCO) framework, where the OCO learner makes a prediction $W_t$ and receives a loss $f_t(W_t)$, where $f_t$ is convex, and gradient $g_t = \nabla_W f_t(W_t)$, which is used to make the next prediction such that the regret is minimized: $R_T(W_1, \ldots, W_T) = \sum_{t=1}^T f_t(W_t) - f_t(W^*)$. $R_T$ measures the closeness of the objective value of the predictions with the optimal comparator $W^* = \arg\min_W \sum_{t=1}^T f_t(W)$. There are online-to-batch conversion techniques (Cesa-Bianchi et al., 2004) which can serve as convergence rate guarantees in stochastic convex optimization. For instance, a regret $R_T$ in OCO is equivalent to a convergence rate $R_T/T$ in stochastic convex optimization. The following theorem establishes a regret upper bound on CASPR algorithm.

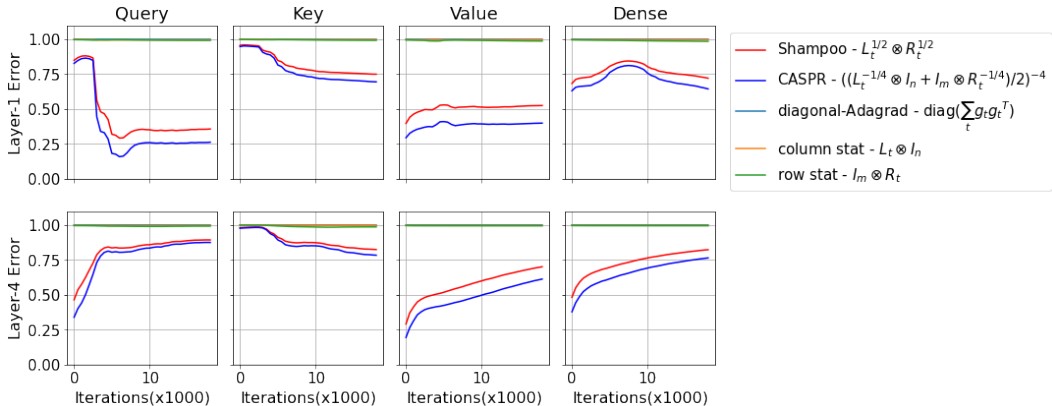

Figure 1: Error plots in a Transformer: $\min_c \|c\hat{H}_t - H_t\|_F / \|H_t\|_F$ (which gives the best approximation error up to a scale $c$ at iteration-$t$). Here $L_t = \sum_t G_t G_t^\top$, $R_t = \sum_t G_t^\top G_t$, where $G_t \in \mathbb{R}^{m \times n}$ denotes the downsampled gradient ($m = n = 128$) of the fully-connected layer, and $H_t = \sum_t g_t g_t^T$, where $g_t = \overline{\text{vec}}(G_t) \in \mathbb{R}^{mn}$. We note that diagonal-Adagrad, column-stat, and row-stat approximations have high relative error $\sim 1$, while the combination of axes statistics $(L, R)$ such as Shampoo, CASPR, approximate full-matrix statistic $H_t$ comparably well during training of a 6 layer Transformer. Here, each row corresponds to a layer/encoder block which constitutes a self-attention sub-layer followed by a MLP sub-layer. We picked the Query, Key, Value transformations from self-attention sub-layer, while the Dense transformation is picked from the MLP sub-layer. More details of this experiment are in Appendix A.2.

**Theorem 4.1** (Regret upper bound of CASPR (Algorithm 1)). *Given that the loss functions $f_t$ : $\mathbb{R}^{m \times n} \to \mathbb{R}$, $\forall t \in [T]$ are convex and $G$-Lipschitz in $\ell_2$-norm i.e., $\|\nabla f_t(W)\|_2 \leq G$, $W \in \mathbb{R}^{m \times n}$, Algorithm 1 incurs the following regret*

$$\sum_{t=1}^{T} f_t(W_t) - f_t(W^*) \leq \sqrt{2r} D \operatorname{tr}\left( \left( (\tilde{L}_T^{-1/4} \otimes I_n + I_m \otimes \tilde{R}_T^{-1/4})/2 \right)^{-2} \right)$$

$$\leq \sqrt{2r} D \operatorname{tr}\left( \tilde{L}_T^{1/4} \otimes \tilde{R}_T^{1/4} \right) = \mathcal{O}(\sqrt{T})$$

*when $\eta = D/\sqrt{2r}$, where $r = \max_t \operatorname{rank}(G_t)$, $D = \|W_t - W^*\|_F$*

The proof is given in Appendix A.3. The above theorem establishes a tighter regret bound for CASPR (first inequality) than the regret upper bound of Shampoo (second inequality). Furthermore, the upperbound $\mathcal{O}(\sqrt{T})$ conveys that Algorithm 1 is asymptotically optimal in the online convex optimization setting. We also discuss existing reductions from regret bound guarantees to non-convex optimization in Appendix A.6.

## 5 EXPERIMENTAL RESULTS

We conduct several experiments to evaluate the performance of CASPR on graph neural networks (Battaglia et al., 2018)and Transformers (Vaswani et al., 2017). We compare CASPR with adaptive regularization methods such as Adam (Kingma & Ba, 2014) and Shampoo (Gupta et al., 2018; Anil et al., 2020). Specifically, we employ blocking Shampoo (Anil et al., 2020) and similarly in CASPR, where large parameters are partitioned into blocks to precondition them separately. CASPR has a computational and memory complexity of $\mathcal{O}(m^3 + n^3)$ and $\mathcal{O}(m^2 + n^2)$ respectively, which is the same as Shampoo, while diagonal preconditioners have time and memory complexity of $\mathcal{O}(mn)$.

For Shampoo and CASPR, inverse root operations are performed once every 20 iterations, and the inverted statistics $L_t^{-1/4}$ and $R_t^{-1/4}$ are reused to amortize the cost of the inverse operations. The inverse fourth root operations are computed via an iterative routine as in Anil et al. (2020) which uses coupled Newton algorithm (Guo & Higham, 2006; Iannazzo, 2006). Given the wide applicability of Shampoo in several benchmarks and existing comparisons with Fisher based methods in Anil et al.

(2020), we make a more thorough comparison with Shampoo. However, the possibility of adapting our method to approximate the Fisher matrix could be a fruitful avenue for future research.

To ensure fair comparison with the aforementioned methods, we incorporate exponential moving averages into Line 4 of Algorithm 1 as follows: $\tilde{L} := \beta_2 \tilde{L} + (1 - \beta_2) G_t G_t^\top, \quad \tilde{R} := \beta_2 \tilde{R} + (1 - \beta_2) G_t^\top G_t$. For more detailed insight into the Shampoo and CASPR settings used in our experiments, see Appendix A.4. We conduct our experiments using the JAX (Bradbury et al., 2018) and Flax (Heek et al., 2023) frameworks. The CASPR code is adapted from the Optax (Babuschkin et al., 2020) implementation of Shampoo (Anil et al.), which is a JAX implementation.

## 5.1 GRAPH NEURAL NETWORK TRAINING ON OGBG-MOLPCBA

Our focus now shifts to evaluating the generalization performance of CASPR on a graph neural network (GNN) (Battaglia et al., 2018). The GNN is trained on the OGBG-molpcba dataset (Hu et al., 2020) using init2winit (Gilmer et al., 2023). The GNN model, implemented using the Jraph library (Godwin* et al., 2020), constitutes 18 fully-connected layers. These layers collectively account for 3.5M parameters with the largest layer having dimensions of $1024 \times 256$. The training data consists of 350,343 graphs and the test set has 43,793 graphs. The nodes are represented as 9-dimensional feature vector and edges are represented as a 3-dimensional feature vector and the task is to predict 128 binary labels each corresponding to a biological activity.

Our training process uses binary cross entropy loss for each class among the 128 with a batch size of 512. We utilize a specific learning rate schedule, involving a linear warmup followed by a cosine decay. In this benchmark, we compare CASPR against Shampoo and AdamW. We use random search with upto 300 hyperparameters, where we search over weight decay, learning rate and momentum parameter for all the algorithms. We fix the dropout to 0.1. The test average precision are depicted in Figure 3 which, demonstrates that CASPR shows relatively 1.4% better test average precision than Shampoo and 7% better test average precision than AdamW. While CASPR and Shampoo are run for 60,000 iterations and take almost the same amount of time (Figure 3), AdamW is calibrated to run for 72,000 iterations such that training finishes in the same amount of time as CASPR or Shampoo. Our data-parallel training runs required 4 TPUv2s, however, the walltime in Figure 3 is computed on TPU v4s to utilize more latest optimizations that TPU v2s don't offer.

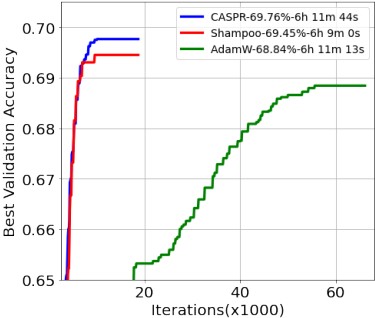

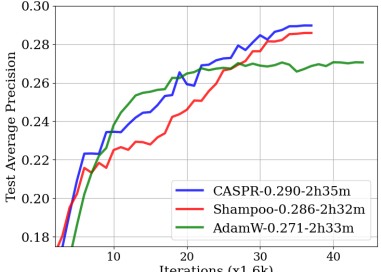

Figure 2: Transformer on Parts of Speech dataset: best validation accuracies reached by CASPR (69.76%), Shampoo (69.45%), AdamW (68.84%). Shampoo and CASPR is run for 18,750 iterations, but AdamW is run for 66,180 iterations so that it takes the same amount of time as CASPR and Shampoo. Shampoo and CASPR take almost the same amount of time, while CASPR gives a much better accuracy.

Figure 3: GNN on OGBG-molpcba dataset: CASPR demonstrates better test average precision (0.29) compared with Shampoo (0.286) and AdamW (0.271). Here Shampoo and CASPR are run for 60,000 steps, however, Adam is run for 72,000 steps, as a result, AdamW takes the same amount of time as Shampoo and CASPR. Here, Shampoo and CASPR take approximately the same amount of time, while CASPR gives a better test AP.

## 5.2 TRANSFORMER ON A UNIVERSAL DEPENDENCIES DATASET

To further verify our generalization performance by using a transformer encoder model (Vaswani et al., 2017) with around 16.6M parameters on a parts-of-speech tagging dataset from universal dependencies (Nivre et al., 2020) with about 202,989 tokens. The largest layer of our model is of size $256 \times 2048$ and a depth of 6 transformer encoder blocks where each block comprises a self-attention mechanism with 8 attention heads and an MLP with hidden dimension 2048. We train our model for

18750 iterations with a batch size of 256, and 48 randomly selected hyperparameter configurations, where we search over weight decay, learning rate and momentum hyperparameters, while fixing the dropout to 0.3, and pick the run which gave best validation error. We use an inverse square root learning rate schedule. In this benchmark, Shampoo reports a best validation accuracy percentage of 69.45%, and AdamW gives a validation accuracy of 68.84%, whereas CASPR reports 69.76%, giving a relative improvement of 1% in error percentage (100-Accuracy) over Shampoo and 3% improvement over AdamW. Figure 2 shows validation performance with similar observations. We run our AdamW baseline for 3.53 times more iterations so that it spends the same training time as Shampoo and CASPR. However, Shampoo and CASPR take about the same training time while they were run for the same number of iterations 18,750. We use one Nvidia A100 gpu for this benchmark.

### 5.3 Auto-regressive Language Modeling

We train a decoder-only GLU based Transformer model (Shazeer, 2020; Vaswani et al., 2017) with 234 million parameters on C4 Dataset (Raffel et al., 2020) with varying batch size (number of sequences per iteration) and sequence length 1024. In Figure 4 we showcase two experiments a) 14 million parameter model with model dimension - 256, hidden dimension - 682 and 8 transformer layers trained with batch size of 8192 sequences with sequence length 1024 for 20,000 iterations totalling $\sim$ 167 billion tokens, and b) 234 million parameter model with model dimension - 1024, hidden dimension - 2730 and 16 transformer layers trained with batch size of 256 sequences with sequences length 1024 for 160,000 iterations totalling $\sim$ 42 billion tokens. Training involved 16 TPU v3s for 234M model with 256 batch size and 64 TPU v3s for 14M model with 8192 batch size, using paxml software (Google). In Figure 4, we compare Shampoo and CASPR's performance where CASPR demonstrates better loss compared to Shampoo. We used linear warmup with cosine decay to zero learning rate for both the optimizers. Given the superior performance of Shampoo and CASPR against Adam in OGBG-molpcba and universal dependencies benchmarks, and due to expensive nature of the language modeling experiments, we omitted Adam from our comparison.

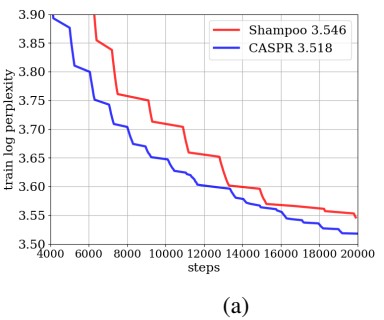
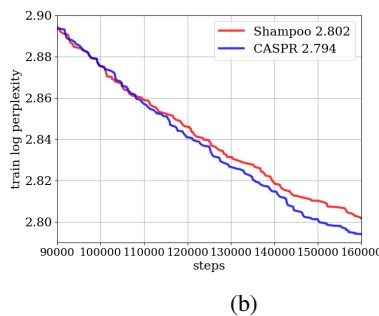

(a)                                                      (b)

Figure 4: GLU based decoder-only Transformer model trained on C4 dataset with varying batch size (number of sequences), number of parameters and total number of steps a) 8192 batch size and 14M parameters for 20,000 steps b) 256 batch size and 234M parameters for 160,000 steps. In both the training runs we see CASPR outperforming Shampoo in log perplexity.

## 6 Conclusions and Future Work

In this paper, we have introduced the CASPR preconditioner for deep neural network optimization that considers the inherent matrix structure of fully-connected layer parameters to construct dedicated preconditioners for each dimension or axis of the parameter. CASPR uses a Kronecker-sum inspired combination to approximate the full-matrix statistic more accurately than existing Kronecker-product based method Shampoo. We establish tighter regret bound guarantees within the online convex optimization framework compared to Shampoo, and demonstrate better training and generalization performance in our deep learning experiments. As future work, CASPR can be adapted to approximate the Fisher matrix in Natural Gradient Descent (Amari, 1998).

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

# A APPENDIX

## A.1 PROOFS

### A.1.1 DERIVATION OF COLUMN PRECONDITIONER

**Lemma A.1.** *Consider the sparsity constraint $\mathcal{S}_{L \otimes I} = \{L \otimes I_n : L \succeq 0 \in \mathbb{R}^{m \times m}\}$ in (3), then the optimal solution $L^*$ for the subproblem (3) (rewritten below) is as follows:*

$$L^* = \arg\min_{L \succeq 0} \|L \otimes I_n - \sum_{s=1}^{t} g_s g_s^\top\|_F = \frac{1}{n} \sum_{s=1}^{t} G_s G_s^\top,$$

*Proof.* Let $P$ is a permutation matrix such that $P \overline{vec}(G) = \overline{vec}(G^\top)$ for any $G \in \mathbb{R}^{m \times n}$, then:

$$
\begin{aligned}
(L \otimes I_n) \overline{vec}(G) &= \overline{vec}(LG) \\
&= \overline{vec}(L(G^\top)^\top) \\
&= \overline{vec}((G^\top L)^\top) \\
&= P^\top \overline{vec}((G^\top L) \\
&= P^\top (I_n \otimes L) \overline{vec}(G^\top) \\
&= P^\top (I_n \otimes L) P \overline{vec}(G)
\end{aligned}
$$

Thus $L \otimes I_n = P^\top (I_n \otimes L) P$ is a block-diagonal matrix upto a permutation, substituting this in the objective function in the lemma gives:

$$
\begin{aligned}
L^* &= \arg\min_{L \succeq 0} \|L \otimes I_n - \sum_{s=1}^{t} g_s g_s^\top\|_F \\
&= \arg\min_{L \succeq 0} \|P(L \otimes I_n)P^\top - \sum_{s=1}^{t} P g_s g_s^\top P^\top\|_F \\
&= \arg\min_{L \succeq 0} \|I_n \otimes L - \sum_{s=1}^{t} \hat{g}_s \hat{g}_s^\top\|_F \\
&= \arg\min_{L \succeq 0} \|\text{block-diag}(L, \dots, L) - \sum_{s=1}^{t} \hat{g}_s \hat{g}_s^\top\|_F \\
&= \frac{1}{n} \sum_{s=1}^{t} (G_s^\top)^\top G_s^\top \quad \text{(Using Lemma 3.1 and } \hat{g}_s = \overline{vec}(G_s^\top)) \\
&= \frac{1}{n} \sum_{s=1}^{t} G_s G_s^\top
\end{aligned}
$$

$\square$

### A.1.2 PROPERTIES OF KRONECKER PRODUCT, KRONECKER SUM AND LOEWNER ORDER

**Lemma A.2** (Eigendecomposition of Kronecker-Sum). *Let $A \in \mathbb{R}^{m \times m}$, $B \in \mathbb{R}^{n \times n}$ be two positive definite matrices with eigenpairs $\{(\lambda_i, u_i) : i \in [m]\}$ and $\{(\sigma_j, v_j) : j \in [n]\}$, where $\Lambda_{i,i} = \lambda_i$ and $\Sigma_{j,j} = \sigma_j$, then the eigenpairs of the matrix $A \otimes I_n + I_m \otimes B$ are $\{(\lambda_i + \sigma_j, u_i \otimes v_j) : i \in [m], j \in [n]\}$*

*Proof.* While this is a standard property, we give its proof here for completeness. First, we show that $u_i \otimes v_j$ is an eigenvector of $A \otimes I_n + I_m \otimes B$:

$$
\begin{aligned}
(A \otimes I_n + I_m \otimes B)u_i \otimes v_j &= (A \otimes I_n + I_m \otimes B)\,\overline{\text{vec}}(u_i v_j^\top) \\
&= (\overline{\text{vec}}(Au_i v_j^\top) + \overline{\text{vec}}(u_i v_j^\top B)) \\
&= \overline{\text{vec}}(\lambda_i u_i v_j^\top + \sigma_j u_i v_j^\top) \quad \text{(since } Au_i = \lambda_i u_i \text{ and } Bv_j = \sigma_j v_j) \\
&= (\lambda_i + \sigma_j)\,\overline{\text{vec}}(u_i v_j^\top) \\
&= (\lambda_i + \sigma_j)u_i \otimes v_j
\end{aligned}
$$

Thus all the $mn$ eigenpairs of $A \otimes I_n + I_m \otimes B$ are $\{(\lambda_i + \sigma_j, u_i \otimes v_j) : i \in [m], j \in [n]\}$  □

**Lemma A.3** (Eigendecomposition of Kronecker-Product ). *Let $A \in \mathbb{R}^{m \times m}$, $B \in \mathbb{R}^{n \times n}$ be two positive definite matrices with eigenpairs $\{(\lambda_i, u_i) : i \in [m]\}$ and $\{(\sigma_j, u_j) : j \in [n]\}$, where $\Lambda_{i,i} = \lambda_i$ and $\Sigma_{j,j} = \sigma_j$, then the eigenpairs of the matrix $A \otimes B$ are $\{(\lambda_i \sigma_j, u_i \otimes v_j) : i \in [m], j \in [n]\}$*

**Lemma A.4** (Geometric mean of commutable matrices (Gupta et al., 2018)). *Let $0 \preceq A_1 \preceq B_1$ and $0 \preceq A_2 \preceq B_2$ and further assume that $A_1 A_2 = A_2 A_1$, $B_1 B_2 = B_2 B_1$, then $A_1^{1/2} A_2^{1/2} \preceq B_1^{1/2} B_2^{1/2}$.*

**Lemma A.5** (standard properties of Loewner order). *Let $A_1 \succeq A_2 \succ 0$ and $B_1 \succeq B_2 \succ 0$, then*

  *(a)* $A_1 \otimes B_1 \succeq A_2 \otimes B_2$

  *(b)* $A_1^{-1} \preceq A_2^{-1}$ and $B_1^{-1} \preceq B_2^{-1}$

  *(c)* $A_1^{1/2} \succeq A_2^{1/2}$ and $B_1^{1/2} \succeq B_2^{1/2}$

### A.1.3 Connections made among CASPR, Shampoo and Full-matrix Adagrad

***Proof of Lemma 3.4.*** Using Lemma A.2 and A.3 for $L_t^{-1/4} \otimes I_n + I_m \otimes R_t^{-1/4}$ and $L_t^{1/2} \otimes R^{1/2}$ respectively, followed by the fact that eigenvalues of $A^p$ are $p$-th powers of eigenvalues of $A$ for any integer $p$.  □

***Proof of Lemma 3.2.*** In this proof, we drop the subscripts for $\tilde{L}_t$ and $\tilde{R}_t$ in the lemma statement. Let $\tilde{L} = U\Lambda U^\top$ and $\tilde{R} = V\Sigma V^\top$, given that $\Lambda \succ 0$ and $\Sigma \succ 0$ (since $\tilde{L} \succeq \epsilon I_n \succ 0$). Using Lemma A.2 the eigenpairs of the general combination $X_t^{caspr}(p)$ in the lemma are $\{(((\lambda_i^{-1/4p} + \sigma_j^{-1/4p})/2)^{2p}, u_i \otimes v_j) : i \in [m], j \in [n]\}$. Note that the eigenvectors are independent of $p$, so eigenpairs of $\lim_{p \to \infty} X_t^{caspr}(p)$ are $\{(\lim_{p \to \infty}((\lambda_i^{-1/4p} + \sigma_j^{-1/4p})/2)^{2p}, u_i \otimes v_j) : i \in [m], j \in [n]\}$. We now analyze the individual limits for each eigenvalue as follows:

$$
\begin{aligned}
\lim_{p \to \infty}((\lambda_i^{-1/4p} + \sigma_j^{-1/4p})/2)^{2p} &= \lim_{p \to \infty} e^{2p \log((\lambda_i^{-1/4p} + \sigma_j^{-1/4p})/2)} \\
&= \lim_{p \to \infty} e^{\log((\lambda_i^{-1/4p} + \sigma_j^{-1/4p})/2)/(1/2p)} \\
&= e^{\lim_{p \to \infty} \log((\lambda_i^{-1/4p} + \sigma_j^{-1/4p})/2)/(1/2p)}
\end{aligned}
$$

As numerator and denominator in the exponent of the last equality evaluate to zero in the limit, L'Hôpital's rule can be used, which gives the following:

$$
\lim_{p \to \infty} \log((\lambda_i^{-1/4p} + \sigma_j^{-1/4p})/2)/(1/2p) = \lim_{p \to \infty} \frac{2(\log(\lambda_i^{-1/4})\lambda_i^{-1/4p} + \log(\sigma_j^{-1/4})\sigma_j^{-1/4p})}{\lambda_i^{-1/4p} + \sigma_j^{-1/4p}}
$$

$$
= \log(\lambda_i^{-1/4}) + \log(\sigma_j^{-1/4}) \quad \text{(Since } \lim_{p \to \infty} \lambda_i^{-1/4p}, \sigma_j^{-1/4p} = 1)
$$

applying exponential function on both sides gives:

$$
\lim_{p \to \infty}((\lambda_i^{-1/4p} + \sigma_j^{-1/4p})/2)^{2p} = (\lambda_i \sigma_j)^{-1/4}
$$

By Lemma A.3, we know that the $((\lambda_i \sigma_j)^{-1/4}, u_i \otimes v_j) : i \in [m], j \in [n]$ are eigenpairs of the Shampoo preconditioner $X_t^{sh} = L_t^{-1/4} \otimes R_t^{-1/4}$.  □

***Proof of Lemma 3.3.***

Showing $\epsilon I_d + \sum_{s=1}^{t} \frac{1}{r}g_s g_s^T \preceq (\tilde{L}_t) \otimes I_n$, We first upperbound individual rank-1 terms $gg^T$ in Loewner order by proceeding as in the Gupta et al. (2017)(proof of Lemma 9). Using the SVD of $g = \overline{\text{vec}}(G) = \overline{\text{vec}}(\sum_{i=1}^{r} \sigma_i u_i v_i^T) = \sum_{i=1}^{r} \sigma_i (u_i \otimes v_i)$, where $r$ denotes the rank of the matrix $G$ gives:

$$gg^T = (\sum_{i=1}^{r} \sigma_i (u_i \otimes v_i))(\sum_{i=1}^{r} \sigma_i (u_i \otimes v_i))^T \preceq r \sum_{i=1}^{r} \sigma_i^2 (u_i u_i^T \otimes v_i v_i^T),$$

$$\prec r \sum_{i=1}^{r} \sigma_i^2 (u_i u_i^T \otimes I_n) \prec r(GG^T \otimes I_n) \tag{4}$$

$$\prec r(I_m \otimes G^T G) \tag{5}$$

where, the first $\preceq$ follows from $(\sum_{i=1}^{r} w_i)(\sum_{i=1}^{r} w_i)^T \preceq r(\sum_{i=1}^{r} w_i w_i^T)$ and (4),(5) use $v_i v_i^T \prec I_n$, $u_i u_i^T \prec I_m$ respectively.

Summing (4) and (5) for gradients $g_t$, across iterations $t \in [T]$ gives, $\epsilon I_d + \sum_{s=1}^{t} \frac{1}{r}g_s g_s^T \preceq (\sum_t G_t G_t^T + \epsilon I_m) \otimes I_n = \tilde{L}_t \otimes I_n$ and $\epsilon I_d + \sum_{s=1}^{t} \frac{1}{r}g_s g_s^T \preceq I_m \otimes (\sum_t G_t^T G_t + \epsilon I_n) = I_m \otimes \tilde{R}_t$ respectively.

Showing $(\epsilon I_d + \sum_{s=1}^{t} \frac{1}{r}g_s g_s^T)^{-1/2} \succ (\tilde{L}_t^{-1/4} \otimes I_n + I_m \otimes \tilde{R}_t^{-1/4})^2/4$:

Expanding the RHS gives

$$(\tilde{L}_t^{-1/4} \otimes I_n + I_m \otimes \tilde{R}_t^{-1/4})^2 = \tilde{L}_t^{-1/2} \otimes I_n + I_m \otimes \tilde{R}_t^{-1/2} + 2 \cdot \tilde{L}_t^{-1/4} \otimes \tilde{R}_t^{-1/4} \tag{6}$$

Now we use $\tilde{L}_t^{-1/2} \otimes I_n \preceq \left(\epsilon I_d + \sum_{s=1}^{t} \frac{1}{r}g_s g_s^T\right)^{-1/2}$, since $X \to X^{1/2}$ is a monotone operator and $A \succ B \implies A^{-1} \prec B^{-1}$.

$$(\tilde{L}_t^{-1/4} \otimes I_n + I_m \otimes \tilde{R}_t^{-1/4})^2 \preceq 2\left(\epsilon I_d + \sum_{s=1}^{t} \frac{1}{r}g_s g_s^T\right)^{-1/2} + 2 \cdot \tilde{L}_t^{-1/4} \otimes \tilde{R}_t^{-1/4}$$

Note that $A = \tilde{L}_t^{-1/2} \otimes I_n$ and $B = I_m \otimes \tilde{R}_t^{-1/2}$ commute. Using Lemma A.4 gives $AB = \tilde{L}_t^{-1/4} \otimes \tilde{R}_t^{-1/4} \preceq \left(\epsilon I_d + \sum_{s=1}^{t} \frac{1}{r}g_s g_s^T\right)^{-1/2}$. Using this in (6) gives the first inequality.

Showing $(\tilde{L}_t^{-1/4} \otimes I_n + I_m \otimes \tilde{R}_t^{-1/4})^2/4 \succeq \tilde{L}_t^{-1/4} \otimes \tilde{R}_t^{-1/4}$:

Using $(A - B)^2 = (A + B)^2 - 4AB \succeq 0$, since $AB = BA$, gives the second inequality.

$\square$

## A.2   SETUP OF APPROXIMATION EXPERIMENT

In Figure 1, we used the attention-based transformer (Vaswani et al., 2017) with 16.6M parameters from Section 5.2, where the details of the transformer are mentioned. Due to memory constraint we cached the gradients once ever 50 iterations when using Adam optimizer, and the cached gradients are used to construct the plot for each approximation in Figure 1. The Query, Key, Value parameters were of size $256 \times 8 \times 32$ (since 8 attention heads), which are flattened to $256 \times 256$ (putting together all the attention heads' parameters), Dense parameter from MLP sub-layer is of size $256 \times 2048$. Since constructing $\sum_t g_t g_t^\top$ can require large amounts of memory, we downsampled Query, Value, Key and Dense parameters to $128 \times 128$ parameter sizes. While Figure 1 mentions errors of column stat, row stat and diagonal-Adagrad to be nearer to 1, close comparison among them is given in Figure 5.

## A.3   REGRET BOUND ANALYSIS OF CASPR

We restate the theorem upper bounding regret here:

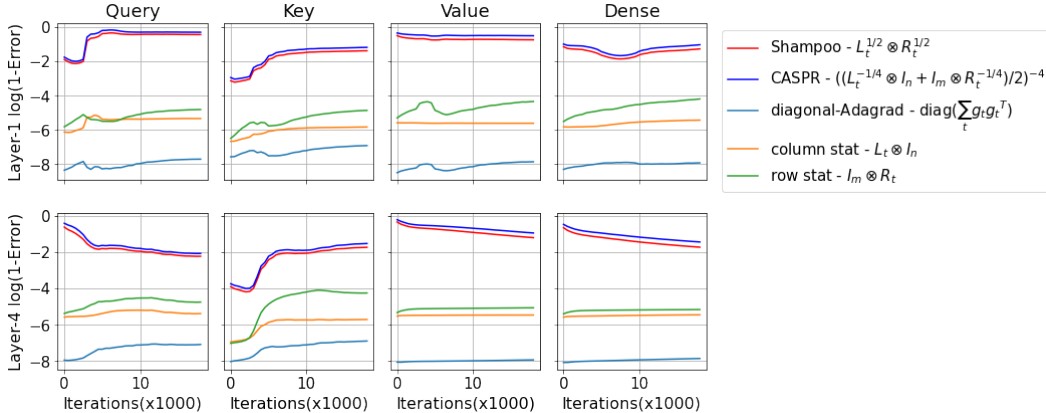

Figure 5: Plot of $\log(1 - \min_c \|c\hat{H}_t - H_t\|_F / \|H_t\|_F)$ in contrast to Figure 1. Here we notice that the column stat and row stat approximate the full-matrix statistic better than the diagonal-Adagrad approximation.

**Theorem A.6** (Regret upper bound of CASPR (Algorithm 1)). *Given that the loss functions $f_t$, $\forall t \in [T]$ are convex, Algorithm 1 gives the following regret*

$$\sum_{t=1}^{T} f_t(W_t) - f_t(W^*) \leq \sqrt{2r}D \operatorname{tr}\left(\left((\tilde{L}_T^{-1/4} \otimes I_n + I_m \otimes \tilde{R}_T^{-1/4})/2\right)^{-2}\right)$$

$$\leq \sqrt{2r}D \operatorname{tr}\left(\tilde{L}_T^{1/4} \otimes \tilde{R}_T^{1/4}\right),$$

*when $\eta = D/\sqrt{2r}$, where $r = \max_t \operatorname{rank}(G_t)$, $D = \max_{t \in \{1,\dots,T\}} \|W_t - W^*\|_F$*

**Proof for Theorem 4.1** (Theorem A.6). We first give a general upperbound to regret $R_T$ by proceeding as in (Hazan et al., 2016).

**Lemma A.7** (A general regret upperbound for adaptive regularization (Hazan et al., 2016)). *Let the regret $R_T = \sum_{t=1}^{T} f(W_t) - f(W^*)$, then*

$$R_T \leq \frac{1}{2\eta}\|w_1 - w^*\|_{X_1^{-1}}^2 + \frac{1}{2\eta}\sum_{t=2}^{T}(w_t - w^*)^\top(X_t^{-1} - X_{t-1}^{-1})(w_t - w^*) + \frac{\eta}{2}\sum_{t=1}^{T}g_t^\top X_t g_t$$

*Proof.* By the parameter update rule from Algorithm 1 we know that $w_{t+1} = w_t - \eta X_t g_t$, where $X_t = ((L_t^{-1/4} \otimes I_n + I_m \otimes R_t^{-1/4})/2)^2$. Subtracting $w^*$ from both sides gives:

$$w_{t+1} - w^* = w_t - w^* - \eta X_t g_t$$

$$\implies \|w_{t+1} - w^*\|_{X_t^{-1}}^2 = \|(w_t - w^*) - \eta X_t g_t\|_{X_t^{-1}}^2$$

$$= \|(w_t - w^*)\|_{X_t^{-1}}^2 + \eta^2 g_t^\top X_t g_t - 2\eta g_t^\top(w_t - w^*)$$

$$\implies g_t^\top(w_t - w^*) = \frac{1}{2\eta}\left(\|(w_t - w^*)\|_{X_t^{-1}}^2 - \|w_{t+1} - w^*\|_{X_t^{-1}}^2\right) + \frac{\eta}{2}g_t^\top X_t g_t$$

where, $\|u\|_A = \sqrt{u^\top A u}$ is a generalized norm defined for positive definite $A$. Summing the above equation over $t \in \{1, \ldots, T\}$ gives the following:

$$\sum_{t=1}^{T} g_t^\top (w_t - w^*) = \frac{1}{2\eta}(\sum_{t=1}^{T}\|(w_t - w^*)\|^2_{X_t^{-1}} - \|w_{t+1} - w^*\|^2_{X_t^{-1}}) + \frac{\eta}{2}\sum_{t=1}^{T} g_t^\top X_t g_t$$

$$= \frac{1}{2\eta}(\|(w_1 - w^*)\|^2_{X_1^{-1}} - \|(w_2 - w^*)\|^2_{X_1^{-1}}$$

$$+ \|(w_2 - w^*)\|^2_{X_2^{-1}} - \|(w_3 - w^*)\|^2_{X_2^{-1}} + \cdots + \|(w_T - w^*)\|^2_{X_T^{-1}} - \|(w_{T+1} - w^*)\|^2_{X_T^{-1}})$$

$$+ \frac{\eta}{2}\sum_{t=1}^{T} g_t^\top X_t g_t$$

$$= \frac{1}{2\eta}(\|(w_1 - w^*)\|^2_{X_1^{-1}} + (w_2 - w^*)^\top(X_2^{-1} - X_1^{-1})(w_2 - w^*) +$$

$$\cdots + (w_T - w^*)^\top(X_T^{-1} - X_{T-1}^{-1})(w_T - w^*)$$

$$- \|(w_{T+1} - w^*)\|^2_{X_T^{-1}}) + \frac{\eta}{2}\sum_{t=1}^{T} g_t^\top X_t g_t \quad \text{(combining 2nd and 3rd terms, 4th and 5th, ...)}$$

$$\leq \frac{1}{2\eta}(\|w_1 - w^*\|^2_{X_1^{-1}}$$

$$+ (w_2 - w^*)^\top(X_2^{-1} - X_1^{-1})(w_2 - w^*) + \cdots + (w_T - w^*)^\top(X_T^{-1} - X_{T-1}^{-1})(w_T - w^*))$$

$$+ \frac{\eta}{2}\sum_{t=1}^{T} g_t^\top X_t g_t$$

The last inequality is because $-\|(w_{T+1} - w^*)\|^2_{X_T^{-1}} \leq 0$. Note that convexity of $f_t$ gives the following:

$$f_t(W_t) - f(W^*) \leq \text{tr}(G_t^\top(W_t - W^*))$$
$$= g_t^\top(w_t - w^*),$$

where $w_t = \overline{\text{vec}}(W_t)$ and $g_t = \overline{\text{vec}}(G_t)$. Using the upperbound on $\sum_t g_t^\top(w_t - w^*)$ derived earlier and the above gives the following:

$$R_T = \sum_{t=1}^{T} f_t(W_t) - f(W^*) \leq \frac{1}{2\eta}(\|w_1 - w^*\|^2_{X_1^{-1}} + \frac{1}{2\eta}\sum_{t=1}^{T}(w_t - w^*)^\top(X_t^{-1} - X_{t-1}^{-1})(w_t - w^*)$$

$$+ \frac{\eta}{2}\sum_{t=1}^{T} g_t^\top X_t g_t$$

$$\square$$

$T_1 = \frac{1}{2\eta}\|w_1 - w^*\|^2_{X_1^{-1}} + \frac{1}{2\eta}\sum_{t=1}^{T}(w_t - w^*)^\top(X_t^{-1} - X_{t-1}^{-1})(w_t - w^*):$

To upperbound this term we first establish a Loewner order on inverse of CASPR preconditioners for consecutive iterations as follows:

**Lemma A.8.** *Preconditioner inverses follow the Loewner order:* $0 \preceq X_1^{-1} \preceq \cdots \preceq X_T^{-1}$

*Proof.* We can expand the CASPR preconditioner as follows:

$$X_t = ((\tilde{L}_t^{-1/4} \otimes I_n + I_m \otimes \tilde{R}_t^{-1/4})/2)^2$$
$$= (\tilde{L}_t^{-1/2} \otimes I_n + I_m \otimes \tilde{R}_t^{-1/2} + 2\tilde{L}_t^{-1/4} \otimes \tilde{R}_t^{-1/4})/4 \quad (7)$$

Note that $\tilde{L}_t = \tilde{L}_{t-1} + G_t G_t^\top \succeq \tilde{L}_{t-1}$ and $\tilde{R}_t = \tilde{R}_{t-1} + G_t^\top G_t \succeq \tilde{L}_{t-1}$, then by Lemma A.5 (a,b,c), $\tilde{L}_t^{-1/2} \otimes I_n \preceq \tilde{L}_{t-1}^{-1/2} \otimes I_n$ and $I_m \otimes \tilde{R}_t^{-1/2} \preceq I_m \otimes \tilde{R}_{t-1}^{-1/2}$. Furthermore, since $\tilde{L}_t^{-1/2} \otimes I_n$

and $I_m \otimes \tilde{R}_t^{-1/2}$ are commutable, by Lemma A.4, we get $L_t^{-1/4} \otimes R_t^{-1/4} \preceq L_{t-1}^{-1/4} \otimes R_{t-1}^{-1/4}$. Putting together the equations, $\tilde{L}_t^{-1/2} \otimes I_n \preceq \tilde{L}_{t-1}^{-1/2} \otimes I_n$, $I_m \otimes \tilde{R}_t^{-1/2} \preceq I_m \otimes \tilde{R}_{t-1}^{-1/2}$ and $L_t^{-1/4} \otimes R_t^{-1/4} \preceq L_{t-1}^{-1/4} \otimes R_{t-1}^{-1/4}$ with (7) gives the following

$$X_t \preceq X_{t-1}$$
$$\implies X_t^{-1} \succeq X_{t-1}^{-1} \quad \text{(By Lemma A.5(b))}$$

$\square$

The above lemma implies that difference between inverses of consecutive preconditioners $X_t^{-1} - X_{t-1}^{-1}$ is positive semidefinite for all $t \in \{1, \ldots, T\}$, thus the individual term

$$(w_t - w^*)^\top (X_t^{-1} - X_{t-1}^{-1})(w_t - w^*) \leq \|(w_t - w^*)\|_2^2 \operatorname{tr}(X_t^{-1} - X_{t-1}^{-1}).$$

Note that trace operator is a norm for positive semidefinite matrices. Summing over all $t \in \{1, \ldots, T\}$ gives

$$
\begin{aligned}
T_1 &= \frac{1}{2\eta} \|w_1 - w^*\|_{X_1^{-1}}^2 + \frac{1}{2\eta} \sum_{t=1}^T (w_t - w^*)^\top (X_t^{-1} - X_{t-1}^{-1})(w_t - w^*) \\
&\leq \frac{1}{2\eta} \left( D^2 \operatorname{tr}(X_1^{-1}) + D^2 \operatorname{tr}(X_2^{-1} - X_1^{-1}) + \cdots + D^2 \operatorname{tr}(X_T^{-1} - X_{T-1}^{-1}) \right) \\
&= \frac{1}{2\eta} D^2 \operatorname{tr}(X_T^{-1}) \\
&= \frac{1}{2\eta} D^2 \operatorname{tr}(((\tilde{L}_T^{-1/4} \otimes I_n + I_m \otimes \tilde{R}_T^{-1/4})/2)^{-2}),
\end{aligned}
$$

where $D = \max_{t \in \{1,\ldots,T\}} \|w_t - w^*\|_2 = \max_{t \in \{1,\ldots,T\}} \|W_t - W^*\|_F$.

$\underline{T_2 = \frac{\eta}{2} \sum_{t=1}^T g_t^\top X_t g_t:}$

To upperbound this term, we use the Loewner order $X_T = ((\tilde{L}_T^{-1/4} \otimes I_n + I_m \otimes \tilde{R}_T^{-1/4})/2)^2 \preceq (\epsilon I_d + \sum_t \frac{1}{r} g_t g_t^T)^{-1/2}$ which gives the following bound:

$$
\begin{aligned}
T_2 &\leq \frac{\eta}{2} \sum_{t=1}^T g_t^\top (\epsilon I_d + \sum_{s=1}^t \frac{1}{r} g_s g_s^T)^{-1/2} g_t \\
&\leq \frac{\sqrt{r}\eta}{2} \sum_{t=1}^T g_t^\top (\epsilon r I_d + \sum_{s=1}^t g_s g_s^T)^{-1/2} g_t
\end{aligned}
\tag{8}
$$

**Lemma A.9** (Gupta et al. (2017)). *Let $g_1, g_2, \ldots, g_T$ be a sequence of vectors and let $M_t = \sum_{s=1}^t g_s g_s^\top$, for $t \leq T$. Given a real-valued function $\phi$ over positive definite matrices, let*

$$H_t = \arg\min_{H \succ 0} \left\{ \operatorname{tr}(M_t H^{-1}) + \phi(H) \right\},$$

*for $t \leq T$, then,*

$$\sum_{t=1}^T \|g_t\|_{H_t^{-1}}^2 \leq \sum_{t=1}^T \|g_t\|_{H_T^{-1}}^2 + \phi(H_T) - \phi(H_0)$$

We proceed as in Gupta et al. (2018) by setting $\phi(H) = \operatorname{tr}(H) + r\epsilon \operatorname{tr}(H^{-1})$ and solving the following reparameterization of the subproblem in the above lemma in variable $X = H^{-1}$:

$$H_t = (\arg\min_{X \succ 0} \left\{ \operatorname{tr}(M_t X) + \phi(X^{-1}) \right\})^{-1}.$$

Using that $\nabla_X (\operatorname{tr}(AX + X^{-1})) = A - X^{-2} = 0 \implies X = A^{-1/2}$ for all $A \succ 0$ and the convexity of the above problem, gives:

$$H_t = \left( r\epsilon I_d + \sum_{s=1}^t g_s g_s^\top \right)^{1/2},$$

By Lemma A.9 and (8):

$$T_2 \leq \frac{\sqrt{r}\eta}{2} \sum_{t=1}^{T} g_t^\top (\epsilon r I_d + \sum_{s=1}^{t} g_s g_s^T)^{-1/2} g_t$$

$$\leq \frac{\sqrt{r}\eta}{2} \left( \sum_{t=1}^{T} g_t^\top (\epsilon r I_d + \sum_{s=1}^{T} g_s g_s^T)^{-1/2} g_t + \mathrm{tr}((r\epsilon I_d + \sum_{t=1}^{T} g_t g_t^\top)^{1/2}) + r\epsilon \, \mathrm{tr}((r\epsilon I_d + \sum_{s=1}^{T} g_s g_s^\top)^{-1/2}) \right)$$

$$= \frac{\sqrt{r}\eta}{2} \left( \mathrm{tr} \left( (r\epsilon I_d + \sum_{t=1}^{T} g_t g_t^\top)(\epsilon r I_d + \sum_{s=1}^{T} g_s g_s^T)^{-1/2} \right) + \mathrm{tr}((r\epsilon I_d + \sum_{t=1}^{T} g_t g_t^\top)^{1/2}) \right) \quad \text{(adding terms 1\&3)}$$

$$= \sqrt{r}\eta \, \mathrm{tr}((r\epsilon I_d + \sum_{t=1}^{T} g_t g_t^\top)^{1/2}))$$

$$\leq r\eta \, \mathrm{tr}(X_T^{-1}) \quad \text{(by Lemma 3.3)}$$

Now setting $\eta = D/\sqrt{2r}$ gives

$$R_T \leq T_1 + T_2 \leq \sqrt{2r}D \, \mathrm{tr}(X_T^{-1}) = \sqrt{2r}D \, \mathrm{tr} \left( \left( (\tilde{L}_T^{-1/4} \otimes I_n + I_m \otimes \tilde{R}_T^{-1/4})/2 \right)^{-2} \right).$$

We also know by Lemma 3.3 and monotonicity of $\mathrm{tr}(.)$ operator, that

$$\mathrm{tr} \left( \left( (\tilde{L}_T^{-1/4} \otimes I_n + I_m \otimes \tilde{R}_T^{-1/4})/2 \right)^{-2} \right) \leq \mathrm{tr} \left( \tilde{L}_T^{1/4} \otimes \tilde{R}_T^{1/4} \right),$$

, thus proving second equality of Theorem 4.1. Given that $f_t$ are $G$-Lipschitz functions, $G_t G_t^\top \preceq G^2 I_m$ and $G_t^\top G_t \preceq G^2 I_n$, thus $\tilde{L}_T \preceq TG^2 I_m$, $\tilde{R}_T \preceq TG^2 I_n$ and $\mathrm{tr}(\tilde{L}_T^{1/4}) \leq m\sqrt{G}T^{1/4}$, $\mathrm{tr}(\tilde{R}_T^{1/4}) \leq n\sqrt{G}T^{1/4}$. Substituting in the above gives $\mathrm{tr}(\tilde{L}_T^{1/4} \otimes \tilde{R}_T^{1/4}) = \mathcal{O}(\sqrt{T})$.

## A.4 HYPERPARAMETER SEARCH SPACES AND OPTIMIZER CONFIGURATIONS

Here we mention the search spaces used for hyperparameter search for graph neural network benchmark 1 and transformer benchmark 2.

| Hyperparameter | Search Space |
|---|---|
| $\eta$ | $[10^{-5}, 0.1]$ |
| $\beta_1$ | $[0.9, 0.999]$ |
| $\beta_2$ | $0.999$ |
| weight decay | $[0.001, 1]$ |

Table 1: Search space for graph neural network benchmark in Section 5.1. $\epsilon$ is set to $10^{-6}$ for Shampoo and CASPR (which are the default values)

| Hyperparameter | Search Space |
|---|---|
| $\eta$ | $[10^{-5}, 0.1]$ |
| $\beta_1$ | $[0.9, 0.999]$ |
| $\beta_2$ | $0.999$ |
| weight decay | $[10^{-4}, 1]$ |

Table 2: Search space for Transformer benchmark in Section 5.2, $\epsilon$ is set to $10^{-8}$ for AdamW and $10^{-6}$ for Shampoo and CASPR (which are the default values)

## A.5 POTENTIAL NEGATIVE SOCIENTAL IMPACT

CASPR tries to reduce the number of iterations required to reach a desirable accuracy, and consequently conserves GPU time. To the best of our knowledge, this poses no negative impact over society.

| Hyperparameter | Search Space |
| --- | --- |
| $\eta$ | $\{10^{-4}, 10^{-3}, 10^{-2}, 10^{-1}\}$ |
| $\beta_2$ | $\{0.99, 0.999\}$ |
| weight decay | $\{0.0, 0.001, 0.1\}$ |
| epsilon | $\{10^{-10}, 10^{-6}\}$ |

Table 3: Search space for 14M parameter language model in Section 5.3. $\beta_1$ is set to 0.9.

| Hyperparameter | Search Space |
| --- | --- |
| $\eta$ | $\{10^{-4}, 10^{-3}, 10^{-2}, 10^{-1}\}$ |
| $\beta_2$ | $\{0.99, 0.999\}$ |

Table 4: Search space of 234M parameter model in 5.3. $\beta_1$ is set to 0.9, weight decay is set to 0.0.

### A.6 REGRET BOUND MINIMIZATION FOR NON-CONVEX OPTIMIZATION

The problem of minimizing smooth non-convex functions $f$ can be reduced to online convex optimization (Agarwal et al., 2019) with sequence of objectives of the form

$$f_t(w) = f(w) + c\|w - w_t\|_2^2$$

where, $c > L$ and $L$- Lipschitz smoothness constant. This method introduced in Agarwal et al. (2019) can be used to obtain convergence guarantees to reach stationary point of the non-convex objective $f$.

Recently, a more direct reduction to regret minimization of linear functions $f_t$ was established in (Cutkosky et al., 2023), to obtain stationary point guarantees in non-smooth non-convex optimization. Thus using regret upper bound guarantees, one can establish stationary point guarantees in the non-convex regime, and hence we focus on the former in the paper.

### A.7 COUPLED NEWTON ITERATION FOR INVERSE P-TH ROOT COMPUTATION

For computing inverses p-th roots we use Algorithm I in (Anil et al., 2020) in Appendix D, originally formulated in Guo & Higham (2006); Iannazzo (2006), which we repeat in Algorithm 2. We limit the number of iterations in while loop to be 100, as done in the JAX implementation of Shampoo. Coupled Newton iteration for inverse 4-th roots has a time complexity of $\mathcal{O}(n^3)$ and memory complexity of $\mathcal{O}(n^2)$. For our experiments, the inverse $p$th root operations were performed using single-precision floating-point format (`float32`). We determined the damping term $\epsilon I$ in Algorithm 1, by scaling the largest eigenvalue of $\lambda_{\max}(L)$ with $\varepsilon$ as outlined in Algorithm 2. This scaling ensures that the modified matrix $\tilde{L} = L + \varepsilon \lambda_{\max}(L)I_m$, maintains an $\ell_2$-condition number not exceeding $1/\varepsilon$. We set $\varepsilon$ to $10^{-6}$ across all our experiments, which caps the condition number of $\tilde{L}$ at $10^6$ which is less than the inverse of machine epsilon for float32 . In the case of the 14M parameter language model showcased in Figure 4a. In that instance, we experimented with $\varepsilon$ values in the set $\{10^{-6}, 10^{-8}\}$, as detailed in Table 3.

### A.8 STANDARD DEVIATION IN EXPERIMENTS

We mention here the standard deviations averaged across 3 seeds for the Transformer on universal dependencies and OGBG-molpcba benchmarks:

| Optimizer | Transformer on Universal Dependencies - Accuracy | OGBG-molpcba - Test mAP |
| --- | --- | --- |
| Shampoo | $69.45 \pm 0.26\%$ | $0.2843 \pm 0.0028$ |
| CASPR | $69.76 \pm 0.14\%$ | $0.2873 \pm 0.0017$ |
| AdamW | $68.28 \pm 0.39\%$ | $0.2701 \pm 0.0011$ |

Table 5: Standard deviations for Transformer on various benchmarks.

### A.9 PRECONDITIONING MULTIDIMENSIONAL PARAMETERS

Our method (as well as Shampoo) can be applied to any parameter tensor. For example, in our experiments with transformers we handle 3D parameters, such as attention layers by flattening the parameter (in the form of higher order tensors) into a 2D tensor by merging the dimensions. The

---

**Algorithm 2** A coupled Newton iteration procedure for computing inverse p-th roots of a PSD matrix, with warm start and singular value projection

---

1: **procedure** MaxSV($G$)
2: **Parameters:** $\varepsilon > 0$, $n_{step}$
3: $v \in \mathbb{R}^n$, where $G \in \mathbb{R}^{n \times n}$
4: $i = 0$, error $= \infty$, $\lambda = 0$
5: **while** $i < n_{step}$ **and** error $> \varepsilon$ **do**
6: $\quad \hat{v} = v / \|v\|$
7: $\quad v = G\hat{v}$
8: $\quad \lambda_{\text{old}} = \lambda$; $\lambda = \hat{v}^\top v$
9: $\quad$ error $= |\lambda - \lambda_{\text{old}}|$; $i = i + 1$
10: **return** $\lambda$
11: **end procedure**
12:
13: **procedure** CoupledIteration($G, p \in \mathbb{N}, X$ (optional))
14: **Parameters:** $\varepsilon > 0$
15: **Outputs:** $G^{-1/p}$
16: $\lambda_{\max} = \text{MaxSV}(G)$
17: $G = G + \varepsilon \cdot \lambda_{\max} \cdot I$
18: $\alpha = -1/p$
19: **if** $X$ is provided **then**
20: $\quad M = X^p G$
21: **else**
22: $\quad z = \frac{1+p}{2\|G\|_F}$
23: $\quad X = \frac{1}{z^\alpha} I$
24: $\quad M = zG$
25: **while** $\|M - I\|_\infty > \varepsilon$ **do**
26: $\quad M_1 = (1 - \alpha)I + \alpha M$
27: $\quad X = X M_1$
28: $\quad M = M_1^p M$
29: **return** $X$
30: **end procedure**

---

attention layers have 3-dimensional parameter $(m, n, k)$ which we precondition by merging the second and third dimensions to a 2-dimensional parameter $(m, n * k)$ and then applying CASPR. For higher dimensional parameters, one can apply a similar transformation by merging more dimensions.

