# OpenReview forum: "Combining Axes Preconditioners through Kronecker Approximation for Deep Learning"
_ICLR.cc/2024/Conference — ICLR 2024 poster_

### Official Review · Reviewer_Vz7y · 2023-10-26

**Soundness:** 4 excellent
**Presentation:** 4 excellent
**Contribution:** 4 excellent
**Rating:** 8
**Confidence:** 5

**Summary:**

This work proposes an optimization algorithm called CASPR that yields tighter convergence guarantees for stochastic optimization. It is inspired from the commonly known Shampoo optimizer (which is a Kronecker product based optimizer) and uses the Kronecker sum approximation to obtain better convergence rate with similar complexity as in the Shampoo case. The technique is well motivated and provides theoretical/experimental proofs/explanations for all the claims in the paper (convergence guarantees, basic properties of kronecker product/sum and Loewner order).

**Strengths:**

1. clearly explains the motivation of a Kronecker sum based precondition, the differences in the updates of CASPR and Shampoo and comparison with diagonal Adagrad statistics in Figure 1 (all in section 3)
2. provides convergence guarantees
3. experiments are performed on transformers and graph neural network tasks (commonly known as difficult tasks) against Shampoo and AdamW, the main competitor optimizers in the literature
4. comprehensive appendix

**Weaknesses:**

1. the paper does not have experiments on computer vision tasks, such as ResNets, ViTs or other important architectures used for benchmarking in the literature

**Questions:**

In the evaluation you introduce a momentum for $L$ and $R$ matrices for a fair comparison against the other optimizers, can you please say what is the impact of momentum terms in this case? Have you experimented without momentum?

I would like to point out some typos in the manuscript:
- at page 4, Lemma 3.1, the Frobenius norm should be squared (the 2nd power is missing from the norm)
- at page 5, when you rewrite the preconditioner $X_t^{caspr}(1)$ in terms of $X_t^L$ and $X_t^R$, I belive the matrices $L$ and $R$ are missing a tilde, based on the definitions of the $X_t^{L/R}$ terms 4 lines above

---

> ### Author Response · Authors · 2023-11-20
>
> Thank you for your valuable review.
> > In the evaluation you introduce a momentum for L and R matrices for a fair comparison against the other optimizers, can you please say what is the impact of momentum terms in this case? Have you experimented without momentum?
>
> We conducted an additional experiment without momentum. We used the optimal hyperparameter for CASPR and removed momentum (i.e, equivalent to beta2 tends to 1) for L and R for CASPR, and noticed that the performance is poor. This could be because in practice momentum is helpful as the curvature could change rapidly. We will conduct a full experiment with several hyperparameters in the next revision.
>
> > at page 4, Lemma 3.1, the Frobenius norm should be…
>
> Thank you for finding this error, we corrected it.
>
> > at page 5, when you rewrite the preconditioner…
>
> Yes, this $ \frac{(L^{-1/2}\otimes I_n + I_m \otimes R^{-1/2})/2 + L_t^{-1/4} \otimes R_t^{-1/4}}{2}$ should be $\frac{(\tilde{L}^{-1/2}\otimes I_n + I_m \otimes \tilde{R}^{-1/2})/2 + \tilde{L}_t^{-1/4} \otimes \tilde{R}_t^{-1/4}}{2}$. We edited this.
>
>
>
> > the paper does not have experiments on computer vision tasks, such as ResNets, ViTs or other important architectures used for benchmarking in the literature
>
> We will try to conduct more experiments on vision tasks in the next revision.

---

> ### Comment · Reviewer_Vz7y · 2023-11-20
>
> It would be great if you could include some large scale Computer Vision experiments, I believe it would add a lot of value to the paper.
>
> Regarding your observation about momentum, I guess $\beta_2 \rightarrow 1$ would result in not incorporating any second order information into the preconditioners $L$ and $R$. Can you please clarify that, I guess you refer to preconditioner updates as they appear in algorithm 1, line 4 (without any $\beta_2$ parameter), right?

---

> ### Author Response · Authors · 2023-11-20
>
> By $\beta_2 \rightarrow 1$, we mean to update $L_t$ and $R_t$ as $L_t = L_{t-1} + G_tG_t^{\top}$ and $R_t = R_{t-1}+ G_t^{\top}G_t$, so the entire history of gradient second order information is accumulated  in $L_t$ and $R_t$ without exponential moving average. Please let us know if we understood your suggestion correctly.

---

### Official Review · Reviewer_gGzL · 2023-11-05

**Soundness:** 3 good
**Presentation:** 3 good
**Contribution:** 3 good
**Rating:** 6
**Confidence:** 3

**Summary:**

The paper proposes an alternative to the recently successful Shampoo optimizer that is derived by minimizing the error of the approximation. They show that their method obtains a faster theoretical convergence guarantee in an online learning setting than the original Shampoo optimizer, and also in practice performs better.

**Strengths:**

- clear development of the method and comparison to Shampoo
- theoretical development that exists of Shampoo extended to caspr
- experiments suggest that caspr improves over shampoo and potentially also other baselines (only compared to Adam(W))

**Weaknesses:**

- abstract and introduction could be written more clearly, it is only mentioned what is achieved in the abstract and introduction but not how this will be done
- relatively limited benchmarking to other methods andmissing error bars on the experiments, it seems like a single training run is reported in the figures. This is the major potential weakness I see in the paper and I am worried that the results are not reproducible. See my questions below (Are the figures created from multiple runs averaged? What is the target for tuning hyperparameters if the plots show validation performance?).
- seems unclear how to extend this method to layers that are not fully-connected

### Minor
- first page sentence ... "..to be at one end of this sequence" confusing

**Questions:**

- On page 3 it says the second-moments are computed per row of the gradient. Is it correct to say that this means it is a neuron-level conditioner? This is something that seems to exist in the context. See for example, Fig 1 "unit-wise" preconditioning in https://arxiv.org/pdf/2305.04684.pdf.
- Are the 300 hyperparameters optimized for each method or only for caspr? What is the target for this hyperparameter tuning? Validation accuracy is the performance reported so is the grid search optimizing that value?
- Why are there no error bars included? Could the experiments be run on at least 3 seeds? Without a single rerun, the results could be random.

---

> ### Author Response · Authors · 2023-11-20
>
> Thank you for your valuable review.
>
> > abstract and introduction could be written more clearly, it is only mentioned what is achieved in the abstract and introduction but not how this will be done
>
> Thank you for pointing this out, we will add more details regarding our method in the introduction of the next revision.
>
> > On page 3 it says the second-moments are computed per row of the gradient. Is it correct to say that this means it is a neuron-level conditioner? This is something that seems to exist in the context. See for example, Fig 1 "unit-wise" preconditioning in https://arxiv.org/pdf/2305.04684.pdf.
>
> Yes, preconditioning each row individually can be imagined as a neuron-level preconditioner. We will cite the mentioned paper in the final draft. Specifically, this originates from a block-diagonal approximation $S_{R} = \{ blockdiag(R_1,R_2\ldots,R_m): R_i\succeq 0\in R^{n\times n},\forall i\in[m]\}$ of full-matrix statistic. Since maintaining this approximation and computing the corresponding preconditioner is infeasible in terms of memory and compute, we additionally develop a statistic/preconditioner following the constraint $S_{I\otimes R} = \{ blockdiag(R,R\ldots,R): R \succeq 0\in R^{n\times n}\}$ in Lemma 3.1, where the additional constraint is that all the blocks $R_1=R_2=\cdots=R_m=R$ are the same. Our work can be seen as extending such neuron level preconditioners by combining them through kronecker approximations. In practice this does give us a big gain as seen in our experimental results.
>
>
> > Are the 300 hyperparameters optimized for each method or only for caspr? What is the target for this hyperparameter tuning? Validation accuracy is the performance reported so is the grid search optimizing that value?
>
> For ogbg-molpcba, we use 300 hyperparameters for Shampoo and AdamW and 200 for CASPR randomly sampled from the search space in Table 1 (Appendix A.4). The hyperparameter configuration which gives the best generalization performance is picked and reported individually for each method.
>
> > Why are there no error bars included? Could the experiments be run on at least 3 seeds? Without a single rerun, the results could be random.
>
> We mention here the standard deviations averaged across 3 seeds for the Transformer on universal dependencies and OGBG-molpcba benchmarks:
>
>
> |Optimizer|Transformer on Universal dependencies - Accuracy|OGBG-molpcba- Test mAP|
> |---------|------------------------------------------------|----------------------|
> |Shampoo  |69.45 +/- 0.26%                                 |0.2843 +/- 0.0028     |
> |CASPR    |69.76 +/- 0.14%                                 |0.2873 +/- 0.0017     |
> |AdamW    |68.28 +/- 0.39%                                 |0.2701 +/- 0.0011     |
>
> We will try to add similar numbers for our language modeling benchmarks in the next revision, as they are more time consuming.
>
>
> > seems unclear how to extend this method to layers that are not fully-connected
>
> Our method (as well as Shampoo) can be applied to any layer. For example, in our experiments with transformers we handle 3D parameters, such as attention layers by flattening the parameter (in the form of higher order tensors) into a 2D tensor by merging the dimensions.  The attention layers have 3D parameter (m,n,k) which we precondition by merging the second and third dimensions to a 2D parameter (m, n*k) and then applying CASPR. For higher dimensional parameters, one can apply a similar transformation by merging more dimensions.
>
> > first page sentence ... "..to be at one end of this sequence" confusing
>
> Thank you for pointing this out. Here, we added a forward reference to Lemma 3.2 to address the confusion.

---

> > ### Author Response · Authors · 2023-11-20
> >
> > > abstract and introduction could be written more clearly, it is only mentioned what is achieved in the abstract and introduction but not how this will be done
> >
> >
> > We made changes to the abstract:
> >
> > The Kronecker-sum based combination allows us to show that CASPR is ordered between a well-known Kronecker product based combination, Shampoo, and full-matrix Adagrad preconditioners in Loewner order, as a result, it is nearer to full-matrix Adagrad than Shampoo.
> >
> > and introduction:
> >
> > CASPR preconditioner is carefully picked from the proposed set of combinations such that it is ordered (sandwiched) between Shampoo and full-matrix Adagrad in Loewner order [1], as a result, it is nearer to the more powerful full-matrix Adagrad preconditioner than Shampoo in Loewner order. This is acheived by utilizing the simple inequality in Loewner order: $(A+B)^2 \succeq 4AB$ for commutable and positive definite $A$ and $B$.  Surprisingly, we show that the established Loewner order also helps us derive tighter regret bounds in the online convex optimization framework [2,3] than Shampoo, which translates to convergence guarantees in stochastic convex optimization via online-to-batch conversions [4].
> >
> > [1] Karl Löwner. Über monotone matrixfunktionen. Mathematische Zeitschrift, 38(1):177–216, 1934
> >
> > [2] Elad Hazan et al. Introduction to online convex optimization. Foundations and Trends® in Optimization, 2(3-4):157–325, 2016
> >
> > [3] Shai Shalev-Shwartz et al. Online learning and online convex optimization. Foundations and Trends in Machine Learning, 4(2):107–194, 2012.
> >
> > [4] Nicolo Cesa-Bianchi, Alex Conconi, and Claudio Gentile. On the generalization ability of on-line learning algorithms. IEEE Transactions on Information Theory, 50(9):2050–2057, 2004

---

> > > ### Author Response · Authors · 2023-11-22
> > > **Further questions?**
> > >
> > > We are open to providing additional clarification or addressing any further questions the reviewer may have concerning the manuscript.

---

### Official Review · Reviewer_a8ib · 2023-11-05

**Soundness:** 3 good
**Presentation:** 3 good
**Contribution:** 4 excellent
**Rating:** 8
**Confidence:** 3

**Summary:**

In this work, the authors propose a new second-order method inspired by the Shampoo optimizer. The authors show that the proposed method is theoretically grounded and empirically tested.  The authors demonstrate the performance of the proposed method on several large models.

**Strengths:**

This is a solid work. Strong theoretical results and empirical results are provided. I do not check the proofs in the appendices.

**Weaknesses:**

The authors should discuss the time and space complexity of the coupled Newton algorithm used for computing the matrix inverse p-root.
Moreover, numerically stability of the proposed method should be discussed.
For example, how to choose the damping term $\epsilon$? Note that the coupled Newton algorithm could be sensitive to the choice of the damping term.
The proposed method may not work well in a low numerical precision setting such as float-32 or bfloat-16. The authors should explicitly discuss this point.

**Questions:**

See the weakness section

---

> ### Author Response · Authors · 2023-11-14
>
> Thank you for your valuable review. We will add the additional details about coupled Newton algorithm for matrix inverse pth root (Appendix D in [1]) in our final draft. Here we clarify a few questions:
>
> > how to choose the damping term ?
>
> For our experiments, the inverse pth root operations were performed using single-precision floating-point format (float32). We determined the damping term, $\epsilon$, by scaling the largest eigenvalue of $\lambda_{max}(L)$ with $\epsilon'$ as outlined for the Shampoo algorithm in line 16 of Algorithm 1 on page 20 in [1]. This scaling ensures that the modified matrix $\tilde{L} = L + \epsilon' \lambda_{max}(L) I_m$, maintains an $\ell_2$-condition number not exceeding $1/\epsilon’$. We set $\epsilon'$ to $10^{-6}$ across all our experiments, which caps the condition number of $\tilde{L}$ at $10^{6}$, except in the case of the 14M parameter language model showcased in Figure 4a. In that instance, we experimented with $\epsilon'$ values in the set {$10^{-6}, 10^{-10}$}, as detailed in Table 3.
>
>
> > The proposed method may not work well in a low numerical precision setting such as float-32 or bfloat-16. The authors should explicitly discuss this point.
>
> Thank you for catching this limitation. The inverse pth root operations of our experiments were conducted in float32. Nevertheless, conducting inverse pth root operations can be limiting in float32 as accurate inverses can only be obtained for matrices with condition number less than $\sim 10^6$. We leave conducting accurate inverse pth root operations tolerant to low precision as a future work.
>
> [1] Rohan Anil, Vineet Gupta, Tomer Koren, Kevin Regan, and Yoram Singer. Scalable second order optimization for deep learning. arXiv preprint arXiv:2002.09018, 2020.

---

### Meta-Review · Area_Chair_biNB · 2023-12-05

**Metareview:**

Adaptive regularization based optimization methods such as full-matrix Adagrad which use gradient second-moment information have potential for improving the efficiency of deep neural network (DNN) training, but are memory intensive and computationally demanding for large neural nets. The authors develop a technique called Combining AxeS PReconditioners (CASPR), which optimizes matrix-shaped DNN parameters by finding different preconditioners for each mode/axis of the parameter and combining them using a Kronecker-sum based approximation. They claim to show tighter convergence guarantees in stochastic optimization compared to a Kronecker product based preconditioner, Shampoo, which arises as a special case of CASPR. They also conduct experiments suggesting that CASPR approximates the gradient second-moment matrix in full-matrix Adagrad more accurately, and claim that it has improvement in training and generalization performance compared to existing practical adaptive regularization based methods such as Shampoo and Adam in a variety of tasks including graph neural network on OGBG-molpcba, Transformer on a universal dependencies dataset and auto-regressive large language modeling on C4 dataset.

All reviewers have a positive assessment of the paper and recommend acceptance despite raising some concerns which seem to have been addressed. I concur and recommend acceptance.

**Justification For Why Not Higher Score:**

While the paper is interesting and reviews are positive I do not think the work is substantially novel to merit spot light/oral as preconditioning has been used in this context before.

**Justification For Why Not Lower Score:**

All reviewers recommend acceptance.

---

### Decision · Program_Chairs · 2024-01-16

Accept (poster)